# Deletions in Glial Fibrillary Acidic Protein Leading to Alterations in Intermediate Filament Assembly and Network Formation

**DOI:** 10.3390/ijms26051913

**Published:** 2025-02-23

**Authors:** Ni-Hsuan Lin, Wan-Syuan Jian, Ming-Der Perng

**Affiliations:** 1Institute of Molecular Medicine, College of Life Sciences and Medicine, National Tsing Hua University, Hsinchu 30043, Taiwan; nancynihsuan@gapp.nthu.edu.tw (N.-H.L.); syuan06134004@gmail.com (W.-S.J.); 2School of Medicine, College of Life Sciences and Medicine, National Tsing Hua University, Hsinchu 30043, Taiwan

**Keywords:** intermediate filament, glial fibrillary acidic protein, Alexander disease, protein assembly

## Abstract

Glial fibrillary acidic protein (GFAP) is classified as a type III intermediate filament protein predominantly expressed in mature astrocytes. It has the ability to self-assemble into 10 nm filaments in vitro, making it particularly valuable for elucidating the sequences essential for filament assembly. In this study, we created a series of deletion mutants targeting sequences in the N-terminal, C-terminal, and central rod domains to explore the sequences critical for the assembly of GFAP into 10 nm filaments. The impact of these deletions on filament formation was evaluated through in vitro assembly studies and transduction assays conducted with primary astrocytes. Our data revealed that deletions at the carboxy end resulted in abnormalities in either filament diameter calibration or lateral association, whereas deletions at the amino-terminal end significantly disrupted the filament assembly process, particularly restricting filament elongation. Furthermore, we discovered that the filament-forming sequences within the rod domain varied in their contributions to filament assembly and network formation. These findings enhance our understanding of the GFAP assembly process in vitro and provide a detailed mapping of the essential regions required for GFAP assembly. These insights hold significant implications for Alexander disease arising from deletion mutations in GFAP.

## 1. Introduction

Intermediate filaments (IFs) constitute one of the three primary cytoskeletal elements in most eukaryotic cells. Over 65 distinct genes within the human genome have been identified as members of the IF protein family [1], and they exhibit differentiation-dependent and cell-type-specific expression patterns. Based on the structure and sequence homology, IF proteins are classified into five major types. Types I and II are keratins that are typically expressed in epithelial cells. Type III IFs include vimentin, desmin, peripherin, and glial fibrillary acidic protein (GFAP), which are expressed in various cell types. Type IV IFs include either neurofilament triplet proteins or α-internexin, which are expressed mainly in the nervous system. Type V IFs include nuclear lamins, which are important components of the nucleoskeleton of a cell. Compared to microtubules and microfilaments, IFs are a much more flexible cytoskeletal component [2]. This property makes IFs major factors that contribute to the mechanical strength of the cell and provide a dynamic scaffold for the organization of the cytoplasm on a structural and functional level [3].

IF proteins feature a tripartite structure consisting of a highly conserved central rod domain, which is flanked by a more variable N-terminal head and C-terminal tail which are low-complexity domains containing intrinsically disordered regions. The centrally located α-helical rod domain spans approximately 310 amino acids in length, which is characterized by long-range heptad repeats of hydrophobic amino acids, with the first and fourth of every seven residues being apolar [4]. This arrangement creates a hydrophobic seal on the helical surface, facilitating the formation of coiled-coil dimers, which serve as the primary driving force sustaining self-assembly. The sequence identity among all IF proteins is particularly high at both ends of the rod domain, making these sequences the signature features of typical IF proteins [5]. In addition to the heptad repeat pattern, a periodic charge distribution with a nonrandom arrangement of acidic and basic residues within the rod has been noted [6,7]. This charge periodicity is thought to contribute to the surface potentials of the rod, potentially stabilizing associations between coiled-coil dimers and higher-order structures through electrostatic interactions [8,9]. The nonhelical head and tail domains of IFs are variable in size and sequence. This diversity may render the end domains responsible for the variability in IF assembly properties in vivo and in vitro [10]. Because their roles most likely vary among and even within IF types, elucidating the functional significance of the end domains has been challenging.

Purified type III IF proteins can self-assemble into 10 nm filaments in vitro in solutions mimicking physiological pH and salt concentrations. Based on previous studies of vimentin, in vitro assembly of IFs has been proposed to follow a three-stage process [11]. An IF monomer initially forms a parallel dimer and then antiparallel tetramers, which subsequently associate into unit-length filament (ULF). The ULF precursors then anneal longitudinally and compact radially into mature 10 nm filaments. However, the molecular mechanisms underlying IF assembly are still poorly understood, largely owing to a lack of high-resolution structural information on the intact IFs. Recently, a new model accounting for cytoplasmic IF assembly has been proposed based on a cryo-electron microscopic tomography study [12]. Their model revealed assembled vimentin filaments to be composed of five protofibrils, each containing eight vimentin polypeptides. The protofibrils connect laterally through the tail domains, and the head domains form a fiber in the lumen of vimentin filaments. The finding that vimentin filaments can integrate low-complexity domains in the complex helical structure demonstrates that intrinsically disordered regions in IF proteins have an important biological function.

GFAP is a type III IF protein transcribed by a gene comprising nine exons spaced over 10 kb on chromosome 17q21 of the human genome [13,14]. Unlike most IF genes, GFAP undergoes alternative splicing to generate multiple isoforms [15], with the predominant isoform being GFAP-α. In mature astrocytes, GFAP, together with vimentin [16], as well as lesser amounts of nestin [17] and synemin [18], constitute the glial filaments which give the astrocyte its distinct morphology, with multiple extensions of fine processes that contact with other cell types in the central nervous system. The expression of GFAP marks the differentiation of astrocytes, and its upregulation is an important component of reactive gliosis, which has both general as well as disease-specific cellular and molecular features [19].

The high abundance of GFAP and its relatively specific expression in astrocytes suggests an important function. While GFAP-null mice display minimal phenotype when unperturbed [20,21,22,23], overexpression of GFAP resulted in a fatal encephalopathy with accumulations of GFAP in Rosenthal fibers [24]. This unexpected finding led to the discovery that mutations in the GFAP gene cause Alexander disease (AxD) [25]. AxD is a rare and often fatal neurological disorder characterized by reactive gliosis, upregulation of GFAP, and the accumulation of Rosenthal fibers. To date, over 100 mutations have been identified that are causally associated with this disease [26]. While the majority of these mutations are missense mutations affecting single amino acids, several exhibit more complex alterations. These include splice site mutations that lead to significant in-frame deletions through exon skipping [27,28,29], as well as nonsense mutations that result in substantial deletions of GFAP [30]. The implications of these AxD mutations are significant for understanding the mechanisms by which they contribute to AxD and for advancing our knowledge of the biology of IF assembly.

In this study, we constructed a series of GFAP mutants with deletions in the N-terminal, C-terminal, and central rod domains. The direct effects of these GFAP deletions on filament formation were analyzed through in vitro assembly studies and transduction assays utilizing primary astrocytes. Our results provide a detailed examination of the GFAP assembly process and allow us to define the essential regions required for GFAP assembly in vitro. These findings have important implications for AxD resulting from deletion mutations in GFAP.

## 2. Results

### 2.1. Expression and Purification of GFAP

In order to identify the limit sequences necessary for GFAP to assemble efficiently into 10 nm GFAP filaments in vitro, we synthesized a series of GFAP deletion constructs by site-directed mutagenesis to sequentially reduce the physical size of GFAP at the indicated region of the amino-terminal, carboxy-terminal, and central rod domains (Figure 1). For N-terminal deletion variants, the NΔ42 fragment is a GFAP cleavage product likely generated by calpain cleavage at ^41^RM^42^, whereas NΔ74, NΔ118, NΔ178, and NΔ225 are GFAP fragments produced by caspase-6 cleavage at ^75^ELND^78^ [31], ^139^VERD^142^ [32], ^174^QEAD^177^ [33], and ^222^VELD^225^ [34], respectively. For C-terminal deletion variants, CΔ121 GFAP is a truncation mutant caused by a nonsense AxD mutation [30], whereas CΔ92 GFAP is a fragment potentially generated by calpain cleavage at ^337^SLKD^340^. The CΔ207 GFAP is an N-terminal fragment produced by caspase-6 cleavage at ^222^VELD^225^ [34]. Additionally, while ∆Ex4 and ∆Ex5 are GFAP deletion variants resulting from AxD mutations [28,29], ∆Ex3 and ∆Ex6 were experimentally generated and included in this study for comparative purposes.

After overexpression in bacteria, recombinant GFAP was purified to homogeneity by a three-step strategy (Figure 2A). Full-length GFAP and its deletion derivatives were first isolated from solubilized inclusion bodies (Figure 2B,C), which were further purified by liquid chromatography using ion exchange (Figure 2D) and gel filtration (Figure 2E) columns in the presence of 6 M urea. After purification, GFAP was obtained in a high yield (~15 mg/L) and high purity (~95%), as judged by quantitative measurement of the Coomassie blue-stained gels (Figure 2E). Gel filtration chromatography showed that purified GFAP in the presence of 6 M urea and a reducing agent behaved as denatured monomers (Figure 2F,G).

### 2.2. Effects of N-Terminal Deletions on GFAP Assembly In Vitro

The assembly characteristics of full-length GFAP and the N-terminal deletion mutants were first examined in vitro by dialysis-based assembly and visualized by negative staining and electron microscopy (EM). While full-length GFAP (Figure 3A) assembled into 10 nm filaments that were long and uniform in diameter, none of the five N-terminal deletion mutants formed normal filaments in vitro. Filaments formed from the NΔ42 GFAP, missing approximately two-thirds of the nonhelical head domain, were shorter and less uniform in diameter (Figure 3B). Notably, these filaments were not always optimally assembled compared to intact GFAP (Figure 3A), as judged by the significant amounts of unassembled or partially assembled materials in the assembly mixture (Figure 3B, arrowheads). The NΔ74 GFAP, with further deletions in the α-helical rod domain, only formed short fibrous stubs with atypical diameters that were often interconnected at their ends (Figure 3C), indicative of an elongation defect. More extensive deletions in the rod domain produced GFAP mutants with more severe disruption in filament assembly. Thus, filaments formed from NΔ118 GFAP were short and variable in diameter (Figure 3D). Upon close inspection, short rod-like structures that looked like filament pieces associated laterally in an overlapping fashion to form filamentous aggregates (Figure 3D, arrowheads). A similar but much more pronounced disruption was seen for the NΔ178 GFAP (Figure 3E, arrowheads), where filament formation was completely blocked as a consequence of this large deletion. The NΔ225 GFAP did not assemble into any filamentous structures; instead, it formed discrete particles resembling unpolymerized material (Figure 3F, arrows), suggesting inefficient filament assembly. We therefore performed a high-speed centrifugation assay to assess filament assembly efficiency, which sediments assembled filaments into the pellet fraction but leaves unassembled materials in the supernatant fraction. While most of the full-length GFAP sedimented efficiently into the pellet fraction (Figure 3G, lane 2, and Figure 3H), the N-terminal deletion mutants remained mainly in the supernatant fraction (Figure 3G, lanes 3, 5, 7, 9, 11, and Figure 3H). These data suggest that deletion of the N-terminal part of GFAP severely disrupted filament elongation and reduced filament-forming efficiency.

### 2.3. Effects of C-Terminal Deletions on GFAP Assembly In Vitro

To identify the C-terminal sequences that are required for GFAP assembly, we carried out in vitro assembly experiments using a series of C-terminal deletion derivatives (Figure 1). Our previous in vitro studies have shown that tail-less GFAP can still form filaments [35], which exhibited an increased tendency to associate laterally. Here, we focused on examining how progressively longer truncations of GFAP at the C-terminus could affect their assembly properties. The CΔ92 GFAP failed to assemble into normal 10 nm filaments but formed filamentous structures with irregular width and length (Figure 4A, arrowheads). Upon close inspection, branching (Figure 4A, arrows), and twisting (Figure 4A, brackets) in filaments were often seen in the assembly mixtures. The CΔ121 GFAP is unique in that it was the first reported nonsense mutation that caused AxD [30], which resulted in a deletion of about 2/3rds of coil 2B and the entire C-terminal tail. When assembled in vitro, CΔ121 GFAP formed ball-like aggregates, a feature best visualized at low magnification (Figure 4B). Although it is difficult to see the structural detail of the electron-dense aggregates when negatively stained with uranyl acetate, areas containing less aggregated materials can sometimes be found (Figure 4B, arrows). At higher magnification (Figure 4B, inset), paracrystal-like structures with an approximate axial periodicity of 21.2 ± 0.6 nm were clearly seen (Figure 4B, white arrows). That the CΔ121 GFAP can form paracrystals under standard in vitro assembly conditions was one of the most unexpected findings of our in vitro assembly studies. More extensive deletion produced the CΔ207 GFAP, which readily formed large amorphous aggregates (Figure 4C), from which filamentous protrusions can be seen (Figure 4C, arrows). These data suggest that filaments assembled from C-terminal deletion mutants were prone to aggregation, therefore we performed a low-speed sedimentation assay designed to monitor the extent of filament aggregation. With the use of this assay, both CΔ207 (Figure 4D, lane 4) and CΔ121 GFAP (Figure 4D, lane 6) sedimented more efficiently than wild-type (WT) protein (Figure 4D, lane 2), with 91% of CΔ207 and 68% of CΔ121 GFAP being found in the pellet fraction. These data confirm that the assembled filaments of CΔ207 and CΔ121 are more prone to aggregation than the WT protein. Since the formation of electron-dense aggregates makes it difficult to see the structural detail of the assembly products, we examined the preassembly materials formed by CΔ207 GFAP before they aggregated to a significant extent. Under preassembly conditions, full-length GFAP mainly formed unit length filament-like structures, which were ~70 nm in length (Figure 4E, arrows). In contrast, CΔ207 GFAP assembled into very long fibrous structures (Figure 4F) with diameters that were much wider (20–30 nm) than normal filaments (Figure 3A). At higher magnifications, additional abnormalities in filaments could be appreciated. In particular, filaments were frequently knotted (Figure 4F, arrowheads), fused laterally (Figure 4F, brackets), and occasionally formed ring-like structures (Figure 4F, black arrows). Knob-like projections were often seen along the length of some filaments (Figure 4F, white arrows).

### 2.4. Effects of Internal Rod Truncation GFAP Filament Assembly In Vitro

Our interest in identifying the limit sequence necessary for forming GFAP filament extends to the rod domain, as internal truncations within this domain caused by GFAP mutations were associated with AxD [27,28,29]. AxD mutations are thought to result in a toxic gain-of-function effect of the GFAP mutant by disrupting normal filament assembly, leading to abnormal aggregation in the form of Rosenthal fibers [36]. The first reported AxD case caused by rod deletion was a splice site mutation that skips exon 4 (ΔEx4), resulting in an internal truncation of 54 residues spanning amino acids 207 to 260 [28]. Our in vitro assembly studies showed that ΔEx4 mutant formed filaments with atypical structures, including short filament length, irregular width, and roughness of filament surface (Figure 5A). Upon close inspection, additional aberrancies in the filament structure could be appreciated. In particular, filament unraveling (Figure 5A, brackets), as well as filament branching (Figure 5A, arrows) and unassembled materials (Figure 5A, arrowheads) were frequently seen. Because the deletion mutation is heterozygous in patients with AxD, we also assessed the assembly behavior of the deletion variant in the presence of increasing proportions of WT GFAP. When ΔEx4 and WT GFAP were coassembled in a 10:90 ratio, the filaments formed were not dramatically different from those of WT alone. Increasing the proportion of ΔEx4 mutant to 25% in the assembly mixture, however, resulted in filaments that were less uniform and variable in width (Figure 5B). At a 50:50 proportion of WT and ΔEx4 mutant, filaments appeared shorter with an irregular diameter (Figure 5C), and filament unraveling was often seen (Figure 5C, arrowheads).

Another splicing error caused by AxD mutation skips exon 5, resulting in an in-frame deletion of 42 residues spanning amino acids 261–302. When assembled alone, the ΔEx5 mutant formed abnormal filaments with slight irregularities in width (Figure 5D). Most unusually, filaments formed with this mutant had a marked propensity to form aggregates, and compacted balls of filaments were prevalent in the assembly mixtures. Coassembly titrations showed that the ΔEx5 deletion mutant had a more deleterious effect upon WT GFAP assembly. Even the presence of 10% ΔEx5 mutant in the coassembly mixture resulted in filaments that had an increased tendency to form aggregates (Figure 5E). Increasing the ΔEx5 mutant to 25% resulted in more filament aggregation (Figure 5F). At a 50:50 proportion, filaments formed aggregates similar to those made by the ΔEx5 mutant alone. Low-speed sedimentation assay revealed that while 86% of WT GFAP remained in the supernatant fraction (Figure 5G, lane 1), 69% of ΔEx5 mutant sedimented into the pellet fraction (Figure 5G, lane 4), suggesting that ΔEx5 deletion promoted filament aggregation. Increasing the proportion of the ΔEx5 mutant in the coassembly mixture also increased the proportion of GFAP in the pellet fractions (Figure 5G, lanes 6 and 8), consistent with the dominant effect of this deletion mutation. High-speed centrifugation assay confirmed that WT GFAP and ΔEx5 deletion mutant, either alone or in combination at indicated proportions, were present almost entirely in the pellet fraction (Figure 5H, lanes 4, 6, and 8), suggesting that they all assembled efficiently. Table 1 summarizes the filament assembly efficiency of all GFAP deletion mutants.

The finding that both ΔEx4 and ΔEx5 deletion variants formed filamentous structures prompted us to wonder whether deleting other regions of the rod would block filament assembly. Within the rod domain, highly conserved sequences were found in the regions immediately preceding exon 4 and following exon 5 (Figure 6A), suggesting a possible role of these sequences in GFAP assembly. We therefore generated internal truncated GFAP variants, deleting either exon 3 (ΔEx3) spanning amino acids 179–206 and part of exon 6 encompassing amino acids 312–340 of GFAP (ΔEx6) to test whether removal of these consensus sequences could impede GFAP assembly. Surprisingly, we found that the ΔEx3 mutant failed to assemble into 10 nm filaments but formed roundish aggregates that were variable in length and irregular in diameter (Figure 6B), some of which were interconnected end-to-end, forming abnormal structures (Figure 6B, brackets) that were never seen in WT filaments. Similarly, ΔEx6 failed to elongate and compact properly into 10 nm filaments but formed rod-like aggregates with an approximate length of 145.65 ± 73 nm and a diameter of 35.22 ± 10 nm (Figure 6C). These data indicate that filament-forming sequences in the rod domain were not equal with respect to their contributions to filament assembly.

### 2.5. Effect of GFAP Deletions on IF Network Formation and Solubility Properties

To investigate the effects of the deletion mutations on filament organization and network formation, WT GFAP or deletion mutants were transduced by lentiviral infection into primary astrocytes derived from GFAP KO rats, which expressed no endogenous GFAP (Figure 7A). While WT GFAP formed an extensive de novo network of densely packed filaments (Figure 7B), the ΔEx3 mutant formed large inclusion-type aggregates (Figure 7C). This is in contrast to the ΔEx4 mutant, which formed short, needle-like aggregates (Figure 7D). The ΔEx5 mutant formed aggregates composed of circular deposits (Figure 7E, arrows), with small ring-like structures scattered throughout the cytoplasm (Figure 7E, inset). GFAP-containing inclusions were also observed in astrocytes transduced with ΔEx6 deletion mutant in >90% of transduced cells (Figure 7F). These data suggest that the deletion mutants affect the ability of GFAP to form an extended filament network in GFAP-free primary astrocytes.

To test whether GFAP aggregates formed by the deletion mutants in transduced astrocytes had solubility properties different from those of WT GFAP, transduced cells were subjected to a detergent-based serial extraction that solubilized nonaggregated forms of GFAP but retained GFAP aggregates [31]. Analysis of the supernatant and pellet fractions by immunoblotting revealed that whereas 80% of WT GFAP was extracted into the supernatant fraction (Figure 7G, lane 2), the ΔEx3 mutant was more resistant to extraction, with 82% of this mutant remaining in the pellet fraction (Figure 7G, lane 3). Similarly, ΔEx5 (Figure 7G, lane 9) and ΔEx6 (Figure 7G, lane 12) deletion mutants were found almost exclusively in the pellet fraction (Figure 7H), consistent with their sequestration into cytoplasmic aggregates. Although 43% of the ΔEx4 mutant was extracted into the supernatant fraction (Figure 7G, lane 6, labeled S), 57% of this mutant was found in the pellet fraction (Figure 7G, lane 6, labeled P), indicating that the ΔEx4 deletion had also adversely affected the solubility of GFAP. Cotransduction of deletion mutants with increasing proportions of WT GFAP (Figure 7G, lanes 4, 5, 7, 8, 10, 11, 13, and 14) resulted in partition patterns similar to those obtained with transduction of deletion mutant alone (Figure 7G, lanes 3, 6, 9, 12). Notably, cotransduced WT GFAP was more resistant to extraction (Figure 7G, lanes 4, 5, 7, 8, 10, 11, 13, 14) compared with cells transduced with WT GFAP alone (Figure 7G, lane 2). The expression levels of GFAP were analyzed by immunoblotting of the total lysates prepared from GFAP-transduced cells (Appendix A). While no endogenous GFAP was expressed in untransduced astrocytes, cells transduced with each transgene produced a single species of GFAP of the expected size and at roughly comparable levels. Thus, the differences we observed in solubility properties are likely due to the deletion per se rather than elevated GFAP expression levels.

### 2.6. Effects of Deletion Mutants on the Endogenous GFAP Networks in Primary Astrocytes

The assembly behavior of the deletion mutants was also examined in primary astrocytes derived from WT rats (Figure 8A) to test the effects of endogenous GFAP IFs on the aggregation process. The distribution of transduced GFAP in relation to the endogenous GFAP was visualized by double-label immunofluorescence microscopy with the use of the monoclonal anti-hGFAP antibody that preferentially recognized human GFAP and the polyclonal anti-panGFAP antibody that recognized both the endogenous rat GFAP and the transduced human GFAP. While WT GFAP mainly formed filaments that colocalized with the endogenous GFAP network (Figure 8B), most of the cells transduced with the ΔEx3 mutant formed cytoplasmic aggregates, which also collapsed the endogenous GFAP networks in most of the transduced cells (Figure 8C). Similar GFAP-containing aggregates were also observed in astrocytes transduced with ΔEx5 (Figure 8E) and ΔEx6 (Figure 8F) mutants. Transduction of ΔEx4 mutant resulted in the formation of needle-like structures that tended to associate into large aggregates (Figure 8D).

To biochemically assess the solubility properties of the mutant GFAPs, primary astrocytes transduced with either WT GFAP or deletion mutants were extracted, and the resulting supernatant and pellet fractions were analyzed by immunoblotting. While about half of the WT GFAP was extracted into the supernatant fraction (Figure 8G, lane 2, labeled S), all deletion mutants except for ΔEx4 were more resistant to extraction, with 84% of ΔEx3 (Figure 8G, lane 3, labeled P), 90% of ΔEx5 (Figure 8G, lane 5, labeled P), and 95% of the ΔEx6 (Figure 8G, lane 6, labeled P) GFAP being found in the pellet fraction (Figure 8H). Although 38% of the ΔEx4 was extracted into the supernatant fraction (Figure 8G, lane 4, labeled S), 62% remained in the pellet fraction (Figure 8G, lane 4, labeled P). Analysis of the total lysates revealed that cells transduced with either WT GFAP (Figure 8G, lane 2, labeled T) or deletion mutants (Figure 8G, lanes 3–6, labeled T) produced proteins of the expected size at comparable levels, suggesting that the observed alterations in filament organization and solubility properties are likely attributable to the deletion in GFAP itself rather than to increased expression levels.

## 3. Discussion

### 3.1. Impact of GFAP Deletions on Filament Assembly

Among the limited in vitro assembly studies conducted to date [37,38,39], identifying the filament-forming sequences in GFAP has proven challenging, primarily because deletion mutations have typically targeted either the N-terminal head or C-terminal tail domains. In this study, we focused on an in vitro assembly study utilizing three sets of GFAP deletion derivatives: one set comprising N-terminal fragments, another consisting of C-terminal fragments, and a third containing truncated variants with deletion within the rod domain. We established a standardized protocol for the purification of recombinant human GFAP and its truncated forms for subsequent in vitro studies. The detailed protocol provided here serves as a convenient reference for the IF research community, ensuring reliable preparation and purification of GFAP. This protocol offers two notable practical advantages: it is straightforward to execute and can be easily scaled up. We have successfully utilized this protocol to express and purify recombinant human GFAP with both high yield and high purity (Figure 2).

Using these purified GFAPs, we showed that while deletions at the carboxy end caused abnormalities in calibrating filament diameter or lateral alignments, deletion at the amino-terminal end more severely disrupted the filament assembly process and restricted filament elongation in particular. In addition, we found that the filament-forming sequences in the rod domain were not equal with respect to their contributions to filament assembly. Table 2 provides a summary of the effects of GFAP deletions on filament assembly in vitro and IF network formation in astrocytes.

### 3.2. The Roles of GFAP Sequences at the N-Terminal Region in Filament Assembly

In the case of amino-terminal deletions, there is a clear correlation between the extent of the deletion and the severity of the resulting filament phenotype. Specifically, the removal of the first 42 amino acids in the N-terminal head domain led to GFAP filaments that were shorter and exhibited irregular diameters. In addition, a significant amount of unpolymerized material was frequently observed, indicative of inefficient assembly. Similar findings were reported in the in vitro assembly of recombinant mouse GFAP lacking the first 39 amino acids [37]. Further deletions by removing additional amino acids contained within the α-helical domains resulted in the N-terminal deletion mutants failing to form any filamentous structures. In particular, the differences in assembly behavior between the NΔ42 and NΔ74 GFAP suggest a more precise delineation of the amino boundary of the rod domain as a crucial segment for filament formation. Within this region, a significant RP box motif containing arginine and proline residues has been identified as playing a critical role in both the initiation and elongation phases of filament assembly [37]. The RP-box motif, located just upstream of the rod domain of GFAP, has the consensus sequence RLSL-RM-PP, which is also found in vimentin, desmin, and peripherin, indicating its general importance in type III IF assembly [5].

Point mutagenesis studies on vimentin, a structurally related IF protein that coassembles with GFAP [16], have demonstrated that the arginines in the PR box are critical for filament assembly, as their interaction with the rod domain through ionic interactions may serve to appropriately position and stabilize dimers [40]. Additionally, the proline residues within or near the RP-box introduce flexibility into the N-terminal head domain, facilitating both intra- and inter-dimeric interactions with the α-helical rod domains [9]. Previous in vitro studies on vimentin have indicated that the head domain may fold back to interact with coil 1A [40,41], underscoring the dynamic nature of the region spanning amino acids 42–74 of GFAP as essential for regulating the early stages of filament assembly. Our data are consistent with related studies on the functional roles of the N-terminal head region in other type III IF proteins, such as vimentin, peripherin [42], and desmin [43]. In vitro assembly studies showed that the headless versions of these IF proteins were unable to assemble into filaments, confirming that the head domain is essential for IF assembly.

### 3.3. The Roles of C-Terminal Region in GFAP Assembly

While the CΔ92 GFAP is capable of forming filament-like structures, there is a notable increase in filament twisting and branching, which suggests potential abnormalities in the lateral alignment of protofibrils that may compromise radial compaction. These findings indicate that the tail domain and C-terminal region of the rod are critical for regulating filament diameter and are essential for ensuring filament stability. Our findings align well with a recent cryo-EM tomography study of fully assembled vimentin intermediate filaments [12], which demonstrated that the C-terminal domain plays a crucial role in radial compaction by laterally connecting protofibrils together. This offers an explanation for why compaction defects were often observed in GFAP filaments assembled from tail-truncated GFAP [35] and the defective assembly characteristics of the CΔ92 deletion mutant we reported here.

A previous study reported a case of AxD caused by a nonsense mutation in GFAP [30], resulting in a deletion of 121 amino acids spanning approximately two-thirds of the final 2B helical segment and the entire C-terminal tail domain. Thus, the CΔ121 GFAP represents the most significant alteration caused by the GFAP mutation observed in AxD. We were surprised to discover that the CΔ121 GFAP can form paracrystals, suggesting that the α-helical coiled-coil alignment necessary for its assembly is preserved in the initial 311 residues of GFAP. These paracrystals exhibit a distinct banding pattern characterized by a 22 nm axial repeat, featuring a 20 nm wide lightly staining band separated by an approximately 2 nm wide darkly staining band. The observed light and dark banding pattern suggests a gap-overlap type structure similar to those seen in other IF paracrystals, such as those formed by neurofilament light chains [44] and lamins [45,46,47]. The high protein density in the overlapping regions results in a lighter appearance due to stain exclusion in negatively stained samples, while the lower protein density in the intervening regions appears darker due to increased stain penetration.

The arrangement of CΔ121 GFAP in the paracrystals agrees best with a model in which dimers are organized in an antiparallel manner, with the helix 1B segments being aligned in phase. Under our standard in vitro assembly conditions, CΔ121 GFAP readily formed well-ordered paracrystals, contrasting with the necessity of divalent cations for effective paracrystal formation in the rod domain of GFAP [48] and lamin B2 [47]. Although the precise role of divalent cations in paracrystal formation remains unclear, it is plausible that the packing in GFAP paracrystals reflects a specific arrangement of the GFAP rod tetramer that is favored in the presence of divalent cations.

Paracrystal-like structures may represent a specific subtype of amyloid fibers, as both are ordered aggregates stabilized by distinct secondary structures. Amyloid fibers, typically associated with pathological proteins, are formed by cross-β sheet bundles, whereas paracrystals are formed through coiled-coil alignments of α-helices, as seen with CΔ121 GFAP. Pathological fibers have also been linked to disease phenotypes in other IF proteins. For example, the G62C missense mutation in the keratin-8 head domain results in the formation of amyloid aggregates, which are believed to contribute to alcoholic steatohepatitis of the liver [49]. Comparative studies of the native keratin-8 head domain and the G62C variant have provided compelling evidence that this mutation accelerates self-association of the head domain, highlighting the role of mutations in driving pathological aggregation.

The CΔ207 GFAP, comprising the first 225 amino acids of GFAP, formed large electron-dense aggregates under standard in vitro assembly conditions. A more astonishing observation was that the CΔ207 deletion mutant formed ribbon-like fibrous structures with diameters much wider than normal under preassembly conditions, in which WT GFAP only formed unit-length filaments. This finding suggests that radial compaction was considerably disrupted in the absence of the C-terminal tail domain and the entire coil 2 domain. The notable increase in filament width indicates that the filament diameter may result from either more loosely wound protofibrils due to impaired inter-fibrillar interactions or an increased number of protofibrils within the filaments.

Similar ribbon-like fibrous structures were also observed in the closely related IF protein desmin when its C-terminal half has been deleted [50]. While the structural basis of these fibers remains unknown, recent studies suggest that the head domain may play a role in their formation. In vitro studies demonstrated that the isolated head domain of desmin can self-associate, leading to phase separation into hydrogels composed of labile cross-β polymers [43]. Although it remains unclear whether the head domain of GFAP can self-associate to form cross-β-enriched structures, there is a distinct possibility that it could undergo phase separation. In support of this possibility, our data showed that the C-terminal deletion variants that retained the head domain are capable of forming fibrous structures, whereas the N-terminal deletion variants missing the head cannot.

### 3.4. The Roles of GFAP Segments in the Rod Domain

The region encompassing amino acids 179–206 is partially encoded by exon 3, and its highly conserved nature indicates that it plays a critical role in filament assembly for most IF proteins. Our data demonstrate that the deletion of this region within the 1B domain significantly impaired filament assembly by promoting aggregation, thereby disrupting the normal function of WT GFAP and interfering with its ability to form proper filaments. This dominant-negative effect likely results from aberrant interactions between the deletion mutant and WT GFAP during filament assembly. A plausible explanation for the observed assembly defect is that the deletion destabilizes inter-dimeric interactions, thereby altering the normal filament assembly process. This hypothesis is now supported by the current model of IF assembly [12], which provides an important advance in our understanding of the architecture and assembly of vimentin filaments (Figure 9), and potentially GFAP as well.

The stability of the tetrameric complex is maintained by a knob–pocket packing arrangement [51], which is reinforced by extensive networks of intermolecular hydrogen bonds, salt bridges, and hydrophobic interactions. Deletion of this region would be expected to disrupt the characteristic knobs-and-pocket interactions, leading to the loss of stabilizing forces within both GFAP dimer and tetramer, ultimately impeding filament assembly. In addition, the disruption of the knob-and-pocket interaction may adversely affect the interdimeric interface, destabilizing the essential helical architecture. This destabilization could also result in decreased solubility of ΔEx3 mutant and promote aggregation when expressed in primary astrocytes. These aggregates could compromise IF dynamics, weakening cytoskeletal integrity, disrupting intracellular transport, and impairing key astrocyte functions such as migration and neuronal support. Given that the deletion of this 1B region of GFAP entirely abolishes the filament assembly, it is evident that the rod region encoded by exon 3 is absolutely required for filament formation.

The ΔEx4 mutant is a deletion mutant of GFAP resulting from a splice site mutation that caused the skipping of exon 4 [28]. This mutation leads to an internal deletion of 54 amino acids, encompassing the last 7 amino acids of coil 1B, all of coil 2A, and the first 4 amino acids of coil 2B. A plausible explanation for this assembly defect is that the ΔEx4 deletion interferes with inter-dimeric interactions through the coil 1B domain, thereby destabilizing the resulting filaments. Supporting this hypothesis, crystal structure analysis of GFAP [52] demonstrated that coil 1B contains a hendecad—a heptad extended by four residues (H204–E207 of human GFAP). While this region is not classified as a heptad repeat, it adopts an extended α-helix conformation and contributes to the formation of a continuous coiled-coil structure. Mutations within this region may distort the fundamental helical architecture, disrupting inter-dimeric interactions and leading to aberrant IF formation both in vitro and in cells. The needle-like aggregates formed by the ΔEx4 deletion mutant resemble those previously reported for H204R [53] and E210K [54] missense mutations located within this region, suggesting that perturbations to the coil 1B structure contribute to their formation. This idea is further supported by the 3D structure of intact vimentin IF [12], which is assembled into modular and intertwined helical structures organized into five protofibrils. The complete assembly of a unit-length protofibril requires A11 tetramers formed by inter-dimeric interactions through the 1B domain, highlighting its importance in filament assembly.

The ΔEx5 mutant produced by a splice site mutation that caused an in-frame deletion of the entire exon 5 has been reported in individuals with type I AxD [29]. This mutation leads to an internal deletion of 42 residues, spanning amino acids 261 to 302, located at the beginning of the 2B domain, but its assembly properties have not been previously examined. In vitro studies revealed that filaments formed by ΔEx5 mutant were long and exhibited a strong tendency to aggregate. These findings suggest that the GFAP sequence encoded by exon 5 plays a crucial role in regulating inter-filament interactions, although it does not appear to be essential for filament elongation. In support of these findings, the current model of IF assembly proposes that the A22 tetramer formed by two antiparallel 2B dimers constitutes the outer surface of the filament [12]. Therefore, deletion of the 2B domain encoded by exon 5 could significantly affect inter-filament interactions. Furthermore, deletion of the 2B domain disrupts the heptad repeat structure and shortens the helical bundles, potentially affecting the interaction between the molecular interlocking region due to misalignment of the highly conserved 1A (amino acids 72–86) and 2B (amino acids 363–376) regions of GFAP. As a result, filament assembly does not block completely but becomes destabilized due to impaired alignment and aberrant interactions at the conserved ends of the rod domain.

When expressed in primary astrocytes, the ΔEx5 mutant formed circular deposits that frequently aggregated into large cytoplasmic inclusions. While the structural basis of these circular deposits remains unclear, they may result from aggregation of destabilized filaments or altered interactions with IF-associated proteins. The deletion mutant was also more resistant to biochemical extraction, which is consistent with its sequestration into cytoplasmic aggregates. Collectively, our data indicate that the GFAP segment encoded by exon 5 is vital for modulating higher-order filament interaction in vitro and for organizing the IF networks in astrocytes. This is supported by the structural studies of the crystalized vimentin fragment containing the first half of coil 2 [55], which starts with a hendecad-based parallel helical bundle that extends all the way through the coil 2A and L2 domains. Deletion of the region would be expected to shorten the α-helical bundle, providing a more understandable basis for the ΔEx5 deletion having significant structural consequences.

The ΔEx6 mutant missing part of the 2B domain of GFAP is unable to assemble into 10 nm filaments and instead forms small rod-like structures in vitro. The inability of the ΔEx6 mutant to form filaments indicates that the region encompassing amino acids 312–340, partially encoded by exon 6, is critical for filament formation. Sequence alignment of GFAP with other IF proteins reveals considerable conservation within this region [56]. Crystallization studies on vimentin have demonstrated that this highly conserved region, located in the second half of the coil 2 domain, is part of the molecular interlock region [57]. The atomic model of vimentin IF reveals that in a fully formed protofibril, highly conserved amino acid residues at the start of coil 1A and the end of coil 2B are brought into close proximity along the entire length of the tetramers [12]. This arrangement facilitates interactions and interlocks successive tetramers, ensuing structural stability and filament integrity. These findings are incredibly important because they elucidate why pathogenic mutations associated with AxD [58] and other genetic disorders of IF proteins [59] typically correlate with the most severe manifestations of these diseases. The deletion of this region is expected to disrupt the heptad periodicity and compromise the continuity of the helical coiling, ultimately obstructing filament assembly.

When overexpressed in astrocytes, this deletion mutant could sequester WT GFAP into aggregates, reducing its availability for forming functional filament networks. This was supported by our observations of increased WT GFAP in the pellet fractions (Figure 7G), indicating its incorporation into non-functional aggregates. Excessive GFAP aggregation can also sequester small stress proteins, such as αB-crystallin and HSP27, reducing their availability for essential cellular functions. In addition, GFAP aggregates disrupt the ubiquitin-proteasome system [60,61] and impair the autophagic responses [62], further limiting the ability of astrocytes to clear aggregates. These disruptions create a vicious cycle, promoting aggregate formation and contributing to astrocyte dysfunction. These findings highlight the potential role of GFAP deletions in AxD, where mutant GFAP aggregates disrupt astrocyte function and contribute to disease pathology.

### 3.5. Concluding Remarks and Future Perspectives

Our biochemical assays of filament assembly provide a detailed mapping of the essential regions required for GFAP filament assembly. However, these assays remain limited to low-resolution studies, requiring further exploration of the full structural impacts of most deletion mutants. A recent cryo-EM tomography study provides high-resolution structural data, enabling more precise modeling of vimentin IF assembly, which can be extrapolated to GFAP. The model of fully assembled IFs also highlights how mutations or deletions in GFAP could disrupt normal filament assembly, leading to aggregation. These insights are particularly relevant for understanding the formation of pathological aggregates, such as Rosenthal fibers in AxD. While GFAP and vimentin exhibit certain structural similarities, they also possess distinct functional properties and tissue-specific expression patterns that tailor IF networks to suit their requirements for mechanical strength, flexibility, and dynamic behavior. Further research aimed at elucidating the molecular differentiation of IFs is essential for enhancing our understanding of their roles in human health and disease, as well as for developing innovative therapies targeting IFs.

Our study investigating the structural and functional impacts of GFAP deletions offers critical insights for developing therapeutic strategies. GFAP deletions disrupt IF assembly and promote GFAP aggregation with potential gain-of-function toxicity. Both genetic and pharmacologic interventions could be pursued to address the consequences of GFAP deletion in AxD. For instance, designing small molecules to stabilize GFAP structure, prevent misfolding, and restore filament assembly could help mitigate its toxic effects. Although no such molecules have been identified to date, high-throughput screening of potential stabilizers presents a promising avenue for future drug discovery. In addition, chemical chaperones targeting Rosenthal fibers or paracrystal-like structures could be explored to reduce GFAP aggregation and restore astrocyte function. Since GFAP upregulation is a hallmark of AxD, reducing GFAP levels may alleviate the toxic effects of its accumulation. Antisense oligonucleotides (ASOs) targeting GFAP have demonstrated the ability to reduce its expression to near-null levels, showing both preventative and restorative effects in preclinical models [63]. Attempts are now underway to extend these findings to humans as a formal therapeutic strategy for treating AxD.

## 4. Materials and Methods

### 4.1. Materials

General chemicals used in this study were of analytical grade or higher, sourced from Sigma-Aldrich (St. Louis, MO, USA). Specific chemicals, such as polyethyleneimine solution, β-D-1-thiogalactopyranoside (IPTG), and protease inhibitors, including leupeptin, aprotinin, N-acetyl-leu-leu-norleucinal (ALLN), and phenylmethylsulfonyl fluoride (PMSF), were also obtained from Sigma-Aldrich. An 8 M urea stock solution was prepared using analytical grade urea (Sigma-Aldrich, Cat. #U5128) and deionized with a mixed-bed ion-exchange resin (Amberlyte, Sigma-Aldrich, Cat. #MB-6113). Buffers were prepared as fresh as possible with ultrapure water (Millipore Milli-Q system, Merck Millipore, Burlington, MA, USA). Ionic and non-ionic detergents, including Nonidet P-40 (NP-40), Triton X-100, Tween 20, and sodium deoxycholate, were purchased from Sigma-Aldrich. Reagents for protein electrophoresis, such as 30% Acrylamide-Bis solution, N,N,N′,N′-tetramethylethylenediamine (TEMED), β-mercaptoethanol, ammonium persulfate (APS), sodium dodecyl sulfate (SDS), and Coomassie brilliant blue R250 were obtained from Bio-Rad (Hercules, CA, USA). Nitrocellulose membranes for immunoblotting were sourced from Pall Corporation (Port Washington, NY, USA), and protein concentrations were determined using the bicinchoninic acid (BCA) assay kit (Thermo Fisher Scientific, Waltham, MA, USA).

Reagents for cell culture, including Hank’s Balanced Salt Solution (HBSS), Dulbecco’s Modified Eagle Medium (DMEM), Minimum Essential Medium (MEM), and Phosphate-Buffered Saline (PBS), were of molecular biology grade and purchased from Thermo Fisher Scientific (Waltham, MA, USA). Cell culture media were supplemented with One Shot fetal bovine serum (FBS, Thermo Fisher, Cat. #A5209401, endotoxin < 10 EU/mL) and 1% penicillin-streptomycin (Gibco, Thermo Fisher Scientific, Waltham, MA, USA; Cat. #15140-122). Enzymes, such as 0.25% trypsin solution (Gibco, Cat. #25200056) and DNase I (Sigma-Aldrich, Cat. #DN25, ≥90% purity), used for enzymatic dissociation of brain tissues, were prepared fresh or thawed immediately before use under sterile conditions. Poly-L-lysine hydrobromide (Sigma-Aldrich, Cat. #P4707, molecular weight 70,000–150,000) was used to coat culture dishes. Dissociated cell suspensions were filtered using 70 µm nylon cell strainers (Corning Inc. Corning, NY, USA; Cat. #431751). Culture dishes and plates were sourced from Greiner Bio-One (Kremsmünster, Austria). DAPI (4′,6-diamidino-2-phenylindole, Sigma-Aldrich, Cat. #D8417) was used for the nuclear staining of cultured cells.

### 4.2. Expression Plasmid Construction

Standard molecular cloning strategies were conducted to generate expression plasmids for all recombinant proteins with indicated deletions (Figure 1). Specific nucleotide changes were introduced into the cDNA of human GFAPα [64] by site-directed mutagenesis using InFusion HD Plus Cloning System (Takara Bio, Mountain View, CA, USA). All newly constructed vectors containing the desired mutations were confirmed by Sanger’s DNA sequencing before use.

### 4.3. GFAP Purification and Molecular Weight Determination

A pET23b expression vector containing either human GFAPα or its deletion mutants was transformed into Novagen’s *E. coli* BL21 (DE3) pLysS strain (Agilent, Santa Clara, CA, USA). Bacteria were grown at 37 °C with continuous shaking at 220–230 rpm until an optical density at 600 nm (OD^600^) reached at least 0.2. GFAP overexpression was induced by the addition of 500 μg/mL IPTG, followed by 4 h of growth under the same conditions. Bacteria were harvested by centrifugation and stored at −80× *g* °C until protein purification. The bacterial pellet was resuspended and homogenized in 50 mL TEN buffer (50 mM Tris pH 8.0, 5 mM EDTA, and 150 mM NaCl supplemented with 0.2 mM PMSF added fresh), followed by centrifugation at 15,000× *g* for 30 min at 4 °C. The resulting pellet was further extracted with Triton buffer (1% (*v*/*v*) Triton X-100 in TEN buffer), followed by high salt buffer (1.5 M KCl and 0.5% (*v*/*v*) Triton X-100 in TEN). After centrifugation, the final pellet, consisting predominantly of GFAP-containing inclusion bodies, was extracted in urea buffer (6 M urea, 20 mM Tris-HCl, pH 8, 5 mM EDTA, 1 mM PMSF) at 4 °C overnight. The whole procedure of sample preparation for subsequent protein purification is summarized in Figure 2B.

After centrifugation at 80,000× *g* for 20 min, the urea-soluble fraction was clarified by adding 0.05% (*v*/*v*) polyethyleneimine to precipitate bacterial DNA. After centrifugation, the clarified supernatant of GFAPs was collected and the A260/A280 absorption spectra were measured to ensure they were free from nucleic acid contamination. Protein samples were passed through a 5 mL HiTrap Q anion exchange column (Cytiva, Marlborough, MA, USA) attached to an NGC chromatography system (Bio-Rad, Hercules, CA, USA). GFAP was then eluted from the column using a linear gradient of 0–0.5 M NaCl in the urea buffer over 1 h at a flow rate of 1 mL/min. Fractions containing GFAP were pooled and further purified by a HiPrep Sephacryl S-300 size exclusion column (Cytiva, Marlborough, MA, USA). To determine the molecular weight, GFAP in a running buffer of 6 M urea, 10 mM Tris HC1, pH 8, 5 mM EDTA, and 10 mM β-mercaptoethanol was passed through the same column pumped at a flow rate of 0.2 mL/min, either on its own or in the presence of molecular weight standards (Bio-Rad, Hercules, CA, USA), including bovine thyroglobulin (335 kD), γ-globulin (158 kD), chicken ovalbumin (44 kD), and horse myoglobin (17 kD). Column fractions were analyzed by SDS-PAGE, followed by Coomassie blue staining, and those containing purified GFAP were aliquoted and stored at −80 °C. Protein concentrations were determined by BCA assay using bovine serum albumin (BSA) as a standard.

### 4.4. In Vitro Assembly and Sedimentation Assay

In vitro assembly of GFAP was carried out as described previously [65]. Briefly, purified GFAP diluted to 0.2 mg/mL in 6 M urea in a low-salt buffer (10 mM Tris-HCl (pH 8.0), 5 mM EDTA, and 14.4 mM β-mercaptoethanol) was dialyzed stepwise against 4 M urea for 4 h and then 2 M urea for 4 h in the same buffer at room temperature, followed by dialyzing against the low-salt buffer without urea overnight at 4 °C. Filament assembly was completed by dialyzing against assembly buffer (10 mM Tris-HCl, 50 mM NaCl, 1 mM DTT, pH 7 ± 0.05) at 34–36 °C for 12–16 h. To account for variations in assembly conditions affecting filament quality and aggregation, control reactions with wild-type GFAP were included in all experiments involving deletion mutants. Additionally, all in vitro assembly experiments were conducted in triplicate and repeated using at least two independent preparations of purified protein.

Assembly efficiency was assessed by high-speed sedimentation assay [66]. Briefly, the assembly mixture was layered onto a 0.85 M sucrose cushion in assembly buffer and centrifuged at 80,000× *g* for 20 min at 20 °C. To investigate the effect of GFAP deletions on filament aggregation, assembly mixtures were subjected to a low-speed centrifugation at 1000× *g* for 5 min at room temperature using a benchtop centrifuge (Eppendorf, Hamburg, Germany). The supernatant and pellet fractions were analyzed by SDS–PAGE, followed by Coomassie blue staining. The distribution of GFAP between pellet and supernatant fractions was analyzed using a ChemiDoc MP Imaging System (Bio-Rad, Hercules, CA, USA) and quantified by the ImageLab Software (v. 6.1, Bio-Rad, Hercules, CA, USA). The filament-forming efficiency was determined as the percentage of the total protein present in the pellet.

### 4.5. Transmission Electron Microscopy

Before the application of the sample, formvar and carbon-coated 200 mesh copper grids (Electron Microscopy Sciences, Hatfield, PA, USA) were glow discharged for 45 s at 20 mA using a PELCO easiGlow Glow Discharge Cleaning System (Ted Pella, Redding, CA, USA). Subsequently, an aliquot of assembly mixture (20 μL) was adsorbed for 60 s to glow-discharged carbon support films. Excess liquid on the grids was carefully drained off using a piece of filter paper, and the grids were allowed to air-dry for 30 s. The grids were then washed twice with ultrapure water, followed by staining for 60 s with 1% (*w*/*v*) uranyl acetate, pH 4.5 (Electron Microscopy Sciences, Hatfield, PA, USA). Grids were examined using a Hitach H-7700 electron microscope operating at an acceleration voltage of 80 kV (Hitachi, Tokyo, Japan). Images were captured using a charge-coupled device camera and further processed by Photoshop CC (Adobe Systems, San Jose, CA, USA). To measure filament width and length, at least three randomly selected electron micrographs from each in vitro assembly experiment were analyzed. Filament widths were measured from 12 to 15 randomly chosen filaments, and quantitative analyses were performed using ImageJ software (version 1.54m, National Institutes of Health, Bethesda, MD, USA). Due to the abnormal filament associations observed in some deletion mutants, accurately determining the true filament length was challenging. As a result, lengths were estimated based on representative filaments and expressed as mean ± SD (Table 2). For certain GFAP mutants, the presence of short filaments was inferred from an increased number of free ends of filament visible in the electron micrographs.

### 4.6. GFAP Knockout Rats and Primary Cell Culture

Astrocyte-enriched primary cultures were prepared from either WT or GFAP-knockout rats [63] using standard procedures as described previously [67]. All cells were cultured at 37 °C in a humidified incubator of 95% (*v*/*v*) air and 5% (*v*/*v*) CO_2_. Briefly, cerebral cortices from P2 pulps were dissected in HBSS followed by incubation with 0.25% (*w*/*v*) trypsin at 37 °C for 15 min in the presence of DNase I. Enzyme-treated cortices were then mechanically dispersed by triturating with a Pasture pipette, and dissociated cells were collected by centrifugation at 1000× *g* for 5 min. Following resuspension in plating medium (MEM containing 5% (*v*/*v*) horse serum, 5% (*v*/*v*) FBS, 100 μ/mL penicillin, and 100 μg/mL streptomycin), suspensions were passed through a 70 μm nylon mesh to remove tissue clumps and cell debris. Cells were collected and plated onto poly-D-lysine-coated plates or dishes at 5 × 10^4^ cells/cm^2^, followed by culturing for 10 days with medium change every 2 days.

### 4.7. Lentiviral Transduction and Immunofluorescence Microscopy

Lentiviruses were produced as described previously [67] by transiently co-transfecting GFAP-containing pLEX vector with the psPAX2 packaging (Addgene, Cat. #12260, Cambridge, MA, St. Louis, MO, USA) and pMD2.G envelope (Addgene, Cat. #12259) vectors into 293T cells (Thermo Fisher Scientific, Waltham, MA, USA). Lentiviral infection was performed by incubating lentiviruses with cultured cells in the presence of 8 μg/mL polybrene. At 4 h post-infection, the virus-containing medium was replaced with fresh culture medium.

After lentiviral transduction, cells were fixed in 4% paraformaldehyde (Electron microscopy Science, Hatfield, PA, USA) in PBS. After being permeabilized with 0.2% (*w*/*v*) Triton X-100 in PBS for 5 min, cells were blocked with 10% (*v*/*v*) normal goat serum (Jackson ImmunoResearch Laboratories, West Grove, PA, USA) in PBS for 1 h at room temperature. After several washes with PBS, cells were incubated with primary antibodies (Table 3) at room temperature for at least 1 h, followed by incubation with secondary antibodies conjugated with Alexa Fluor 488 or Alexa Fluor 594 (Thermo Fisher Scientific, Waltham, MA, USA) for 1 h. Nuclei were visualized by staining with DAPI. Cells were examined by visualization with a Zeiss LSM800 laser scanning confocal microscope (Carl Zeiss, Jena, Germany) using either 20× (0.7 NA) Plan-Neofluar or 40× (1.3 NA) Apochromat objective lens. Images were acquired by Zen software (Ver. 2.3), taking 0.5 μm optical sections and processed for figures using Adobe Photoshop CC (Adobe Systems, San Jose, CA, USA). For quantification of GFAP aggregation, several random fields from at least three coverslips were analyzed by visual assessment of the percentage of transduced cells that displayed GFAP-positive aggregates.

### 4.8. Subcellular Fractionation and Immunoblotting Analyses

Cells cultured in 6-well plates were collected and dounce-homogenized in ice-cold RIPA buffer (50 mM Tris pH 8.0, 150 mM NaCl, 1% NP-40, 5 mM EDTA, 0.5% sodium deoxycholate, and 0.1% SDS) containing PMSF and a mixture of protease inhibitors including aprotinin (2 μg/mL), ALLN (10 μM), and leupeptin (5 μg/mL). A small aliquot of homogenates was taken as total cell lysates and the rest was centrifuged at 16,000× *g* for 15 min at 4 °C. The supernatants were taken as RIPA-soluble fraction and the pellets as RIPA-insoluble fraction, which were resuspended and sonicated in TES buffer (20 mM Tris-HCl, pH 7.4, 5 mM EDTA, 1% (*w*/*v*) SDS) in the presence of β-mercaptoethanol. Protein samples prepared from three independent experiments were loaded onto SDS-PAGE gels and analyzed by immunoblotting.

Immunoblotting was conducted using the wet electrophoretic transfer system (Bio-Rad, Hercules, CA, USA) as previously described [68]. Following the electrophoretic transfer of proteins, nitrocellulose membranes were blocked with 3% (*w*/*v*) BSA in Tris-buffered saline with Tween (TBST; 20 mM Tris-HCl, pH 7.4, and 150 mM NaCl, containing 0.1% (*v*/*v*) Tween 20) at room temperature for 1 h. After being washed with TBST, membranes were incubated with primary antibodies (Table 3) at 4 °C overnight, followed by incubation with StarBright Blue 520 and StarBright Blue 700 secondary antibodies, either individually or in combination, for a minimum of 1 h. Signals from non-saturated exposures of the immunoblots were quantified using Image Lab Software (Bio-Rad, Hercules, CA, USA).

### 4.9. Statistical Analysis

All experiments were repeated at least three times unless otherwise stated. All quantitative measurements were presented as mean ± standard deviation (SD). Two-tailed unpaired *t*-tests were used for comparison between the control and experiment groups. For all statistical analyses, the data were considered statistically different if *p* < 0.05.

## Figures and Tables

**Figure 1 ijms-26-01913-f001:**
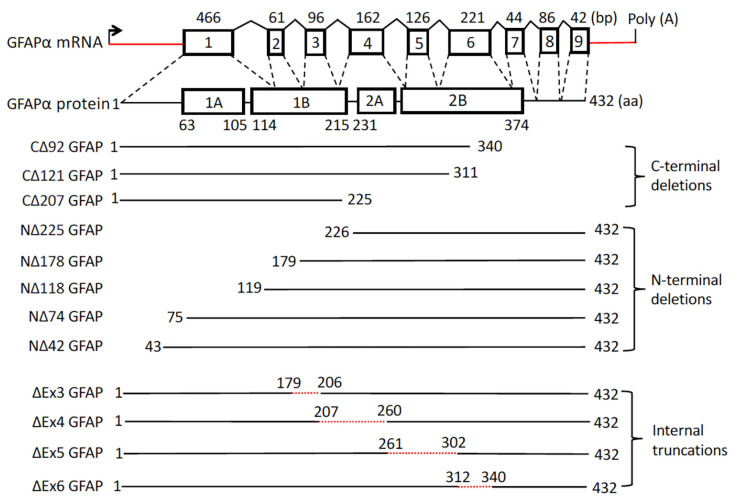
A schematic representation of the GFAP gene structure and its engineered deletion variants. The diagram highlights the gene structure of the major isoform GFAPα, which consists of nine exons (rectangular box). The untranslated regions are marked by red lines (not drawn to scale). The start codon and polyadenylation site (poly (A)) at the end of GFAPα mRNA are indicated. Splicing events are represented by sloped black lines. The sizes of introns and exons are not drawn to scale. bp, base pair; aa, amino acids. The GFAPα protein consists of a central α-helical rod domain, flanked by N-terminal head and C-terminal tail domains (denoted by black bars). Within the rod domain, the heptad repeat-containing segments (coils 1A, 1B, 2A, and 2B shown as boxes) are separated by short linker sequences (L1, L12, and L2 depicted as black bars). The deletion mutations examined in this study are indicated on the left, where NΔX refers to the number of residues deleted from the N-terminus of GFAP, and CΔX refers to the number of residues deleted from the C-terminus of GFAP. GFAP variants with internal truncations were also shown, with the dotted red line indicating the specific locations and number of amino acids deleted within the rod domain.

**Figure 2 ijms-26-01913-f002:**
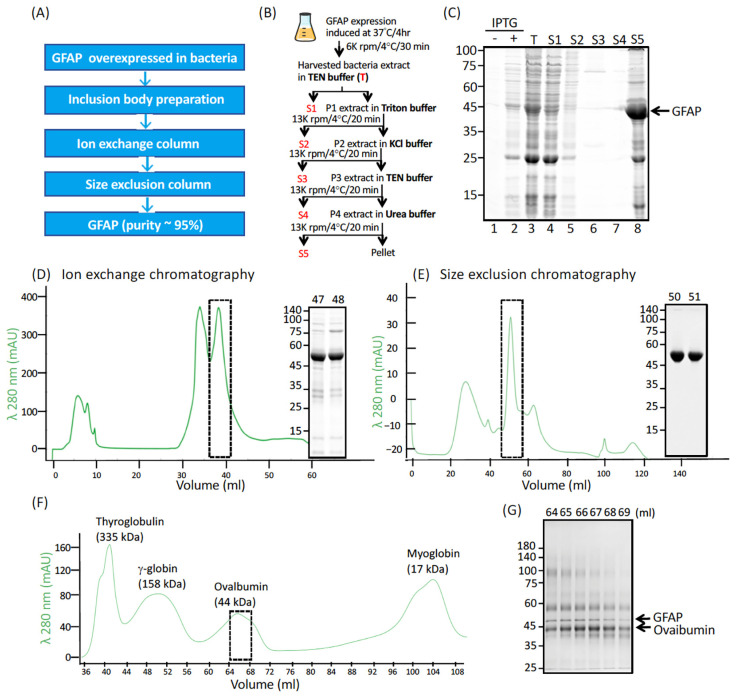
Purification of recombinant GFAP. (**A**) Recombinant GFAP was purified to homogeneity by a three-step strategy. WT human GFAP was used as an example to illustrate the expression and purification of recombinantly produced proteins. (**B**) A schematic diagram showed the sequential extraction protocol used in this study to enrich GFAP in the inclusion bodies. (**C**) Fractions from each step (labeled red in (**B**)) were analyzed by SDS-PAGE followed by Coomassie blue staining. Molecular weight markers (in kDa) are indicated on the left, and GFAP is indicated on the right. GFAP was further purified by ion exchange (**D**) and size exclusion (**E**) column chromatography. (**F**) Determination of molecular weight of GFAP in 6 M urea. Purified GFAP combined with the indicated size exclusion markers in 6 M urea was subjected to size exclusion chromatography. The chromatogram shows the trace of A280 absorbance profile. Selective peak fractions (dashed boxes in **D**–**F**) were analyzed by SDS-PAGE followed by Coomassie blue staining (**G**). Note that most of the GFAP coeluted with ovalbumin (44 kDa).

**Figure 3 ijms-26-01913-f003:**
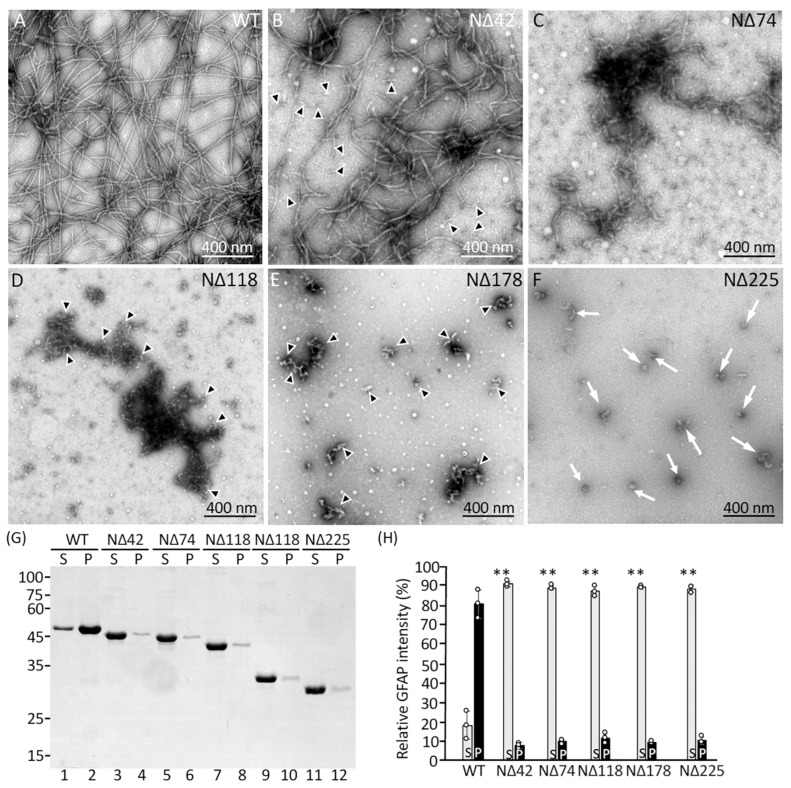
N-terminal deletions disrupted GFAP filament assembly in vitro. Electron micrographs of in vitro assembly structures generated by WT GFAP and N-terminal deletion variants. Purified recombinant WT GFAP (**A**) and the N-terminal deletion mutants NΔ42 (**B**), NΔ74 (**C**), NΔ118 (**D**), NΔ178 (**E**), NΔ225 (**F**) assembled in vitro were negatively stained and visualized by transmission electron microscopy (TEM). Under these assembly conditions, WT GFAP assembled into typical 10 nm filaments with lengths of several microns (**A**). Note that NΔ42 formed filament-like structures with irregular diameters. Some shorter filament pieces that had failed to elongate and compact properly into 10 nm filaments were seen in the background (**B,** arrowheads). Both NΔ74 (**C**) and NΔ118 (**D,** arrowheads) GFAP formed short filament pieces that had a strong tendency to associate laterally into aggregates. Both NΔ178 ((**E**), arrowheads) and NΔ225 ((**F**), arrows) GFAP failed to assemble into extended filaments but formed discrete particles instead. The length of the scale bar is indicated. (**G**) Assembled GFAPs were subjected to high-speed sedimentation assays, and the resulting supernatant (S) and pellet (P) fractions were analyzed by SDS-PAGE followed by Coomassie blue staining. Markers of molecular weight are indicated on the left. (**H**) Quantification of deletion mutants in the pellet fractions was compared to WT controls. Data are mean ± SD from three independent experiments and presented as bar charts. Statistical significance was analyzed by two-tail *t*-test. ** *p* < 0.01. Each white dot represents a biological replicate (*n* = 3).

**Figure 4 ijms-26-01913-f004:**
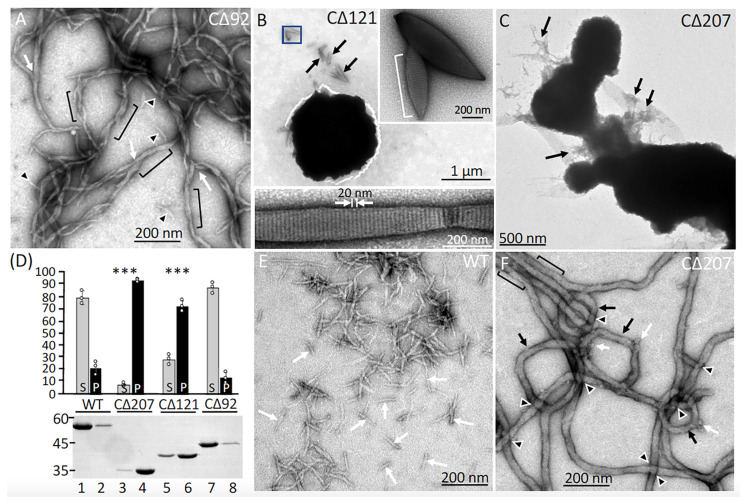
Effects of C-terminal deletion on GFAP assembly in vitro. The C-terminal deletion mutants assembled in vitro were negatively stained and visualized by TEM. Note that filaments formed by CΔ92 GFAP exhibited atypical features, including irregular width, branching ((**A**), arrows) and twisting ((**A**), brackets) in filaments. Unpolymerized materials were often seen in the background ((**A**), arrowheads). The aberrant structures formed by the CΔ92 mutant imply that the tail and the C-terminal part of the 2B domains are critical for regulating filament width and stabilizing filament architecture. The CΔ121 GFAP formed ball-like aggregates, which can be seen at a low-magnification overview (**B**). Sometimes, less aggregated material can be found ((**B**), black arrows), and then, at higher magnification, the paracrystal-like structures that comprise the aggregates are clearly seen ((**B**), inset). The paracrystal (bracket in (**B**), inset) exhibited an approximate axial periodicity of 22 ± 0.6 nm ((**B**), white arrows), featuring alternating light (~20 nm) and dark (~2 nm) banding patterns. The unique structure formed by the CΔ121 mutant suggests that the entire tail domain and the C-terminal 2/3 of the 2B domain play important roles in regulating the assembly and organization of GFAP filaments. The CΔ207 GFAP formed large amorphous aggregates, from which filamentous protrusions can be seen ((**C**), arrows). These aberrant structures indicate uncontrolled assembly of the deletion variants, leading to increased inter-filament interactions. The lengths of the scale bar are indicated. (**D**) Assembled WT GFAP and deletion mutants were subjected to a low-speed centrifugation assay and the resulting supernatant (S) and pellet (P) fractions were analyzed by SDS-PAGE followed by Coomassie blue staining. Quantification of GFAP mutants in the supernatant and pellet fractions was compared to WT controls (**D**). Data are mean ± SD, with each white dot representing a biological replicate (*n* = 3). Statistical significance was analyzed by two-tail *t*-test. *** *p* < 0.01. Under preassembly conditions, WT GFAP mainly assembled into unit-length filaments ((**E**), arrows). In contrast, the CΔ207 GFAP formed cable-like structures with a much wider diameter (**F**). Note that some filaments had closed into ring-like structures ((**F**), black arrows) and aberrant structures, such as knotted filaments ((**F**), arrowheads), lateral fusions ((**F**), brackets), and knob-like projections ((**F**), white arrows) were often observed. These unusual pre-assembly structures indicate that the C-terminal tail domain and the entire coil 2 domain play important roles in the lateral packing and radial compaction of GFAP filaments.

**Figure 5 ijms-26-01913-f005:**
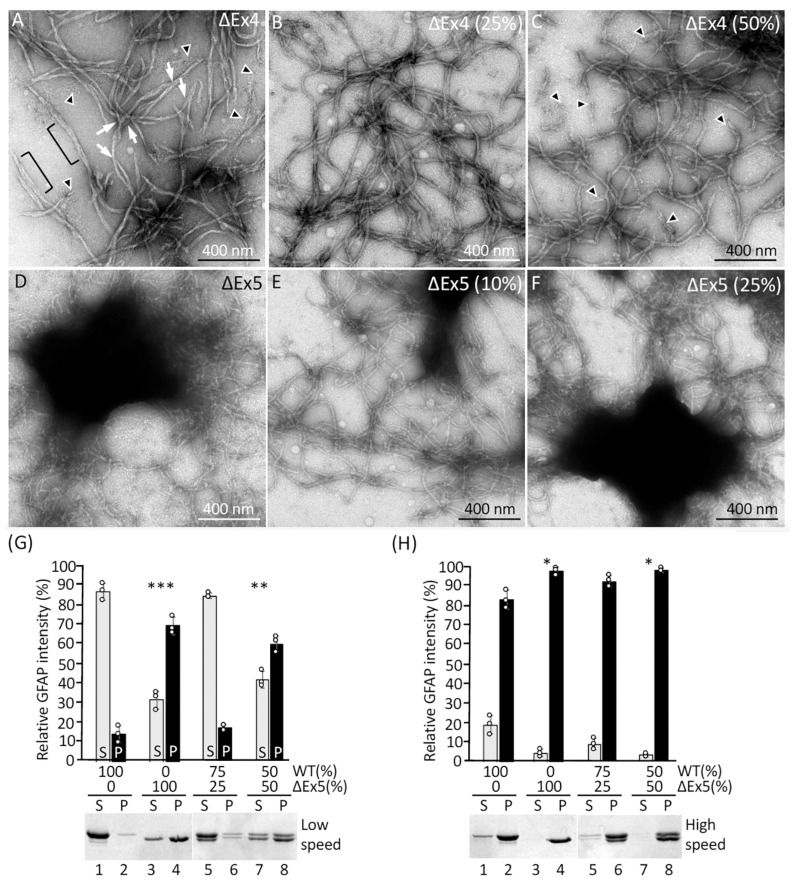
Effects of internal truncation on the in vitro assembly of GFAP. The ΔEx4 mutant alone formed short filaments with atypical characteristics, including protofibrils within filaments ((**A**), brackets), branching ((**A**), arrows), and unraveling ((**A**), arrowheads). The aberrant structures formed by the ΔEx4 mutant indicate the region encoded by exon 4 of GFAP is crucial for stabilizing filament architecture. Coassembly of ΔEx4 mutant with WT protein in 25:75 (**B**) and 50:50 (**C**) proportions resulted in the formation of abnormal filaments (**C**, arrowheads), suggesting that the deletion mutant is dominant over the WT protein. The ΔEx5 mutant alone formed filaments with irregular width and a pronounced tendency to aggregate (**D**), indicating that the sequence encoded by exon 5 of GFAP is essential for regulating inter-filament interactions. Mixing WT GFAP in either 90:10 (**E**) or 75:25 (**F**) proportion with ΔEx5 mutant failed to rescue filament formation and similar aggregates were formed, suggesting the effect of the deletion mutant is dominant over the WT protein. Scale bar represents 400 nm (**A**–**F**). Assembly mixtures were subjected to low-speed (**G**) and high-speed (**H**) sedimentation assays, and the resulting supernatant and pellet fractions were analyzed by SDS-PAGE followed by Coomassie blue staining. Quantification of GFAP in the supernatant and pellet fractions were presented as bar charts. Data are mean ± SD. Each white dot represents a biological replicate (*n* = 3). GFAP levels in (**G**) were analyzed by two-tail *t*-test. Levels of significance are indicated by asterisk (*). * *p* < 0.05, ** *p* < 0.01, *** *p* < 0.001.

**Figure 6 ijms-26-01913-f006:**
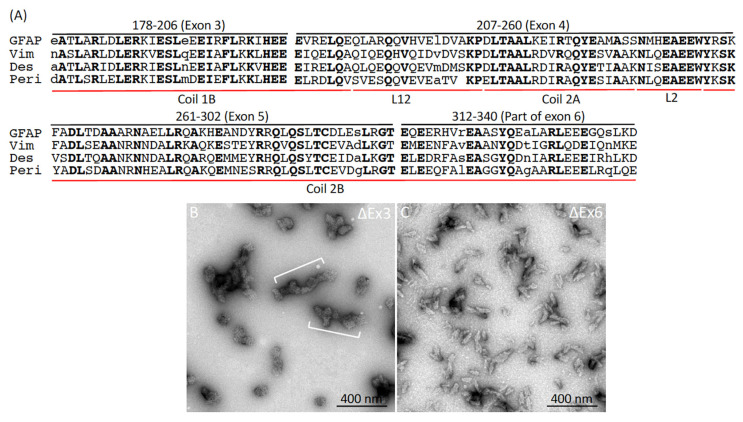
Impact of highly conserved regions within the rod domain on the in vitro assembly of GFAP. (**A**) A comparison of the amino acid sequences of the rod domain, spanning residues 178–340 encoded by exons 3–6 (indicated by black lines), across type III IF proteins. Notably, the majority of amino acids in these regions are highly conserved. Identical amino acids are highlighted in bold, while amino acids unique to each IF protein are shown in lowercase letters. The corresponding helical subdomains (coils 1B, 2A, and 2B) and linker regions (L12 and L2) are indicated by red lines. Under standard in vitro assembly conditions, both ΔEx3 (**B**) and ΔEx6 (**C**) GFAP variants failed to assemble into filaments, instead forming short, rod-like structures that often associated laterally (**B**, brackets).

**Figure 7 ijms-26-01913-f007:**
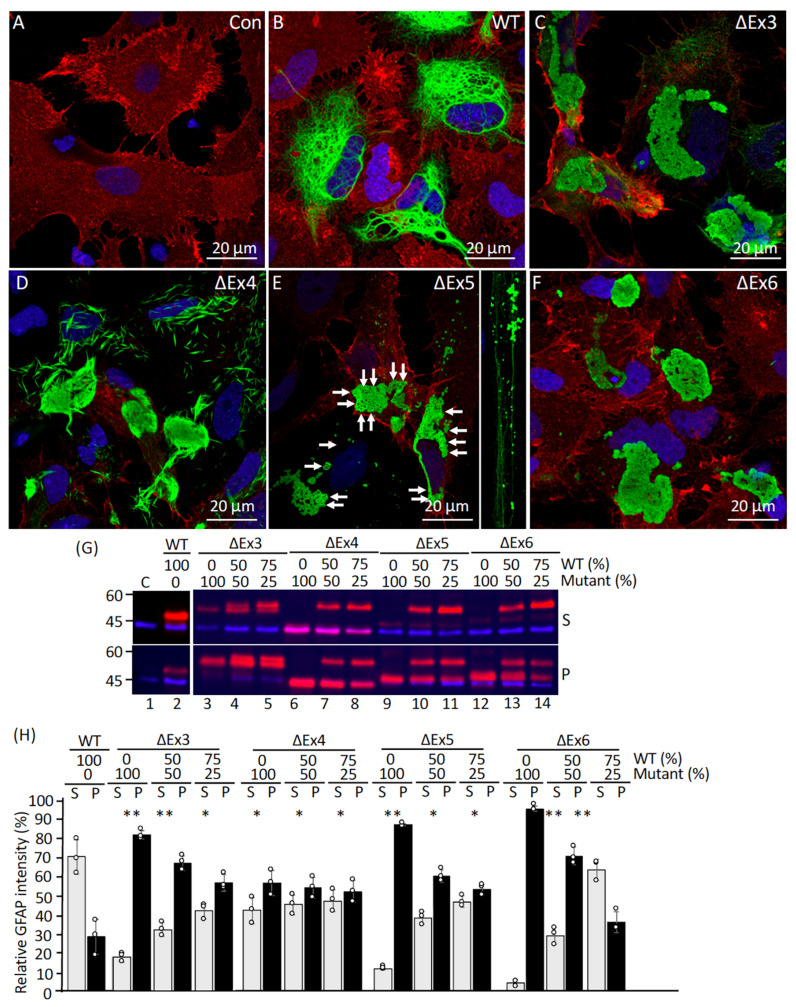
Effect of internal truncation on IF network formation and solubility properties in GFAP-null astrocytes. Primary astrocytes derived from GFAP-null rats were either untransduced (**A**) or transduced with indicated GFAP expression constructs (**B**–**F**). At 72 h after transduction, cells were processed for double-label immunofluorescence microscopy using anti-GFAP (green channel) and anti-aquaporin (red channel) antibodies. Nuclei were visualized by staining with DAPI (blue channel) and merged images were shown. Scale bar represents 20 μm (**A**–**F**). Note that whereas both ΔEx3 (**C**) and ΔEx6 (**F**) GFAP formed large inclusion, ΔEx4 mutant (**D**) formed needle-like structures that were aggregation-prone. The ΔEx5 mutant formed circular deposits ((**E**), arrows) that had a strong tendency to associate laterally into aggregates. The biochemical solubility of GFAP was assessed by extracting untransduced cells ((**G**), lane 1) or cells transduced with WT GFAP ((**G**), lane 2) or deletion mutants either alone or in combination with WT GFAP at indicated proportions ((**G**), lanes 3–14). The resulting supernatant ((**G**), labeled S) and pellet ((**G**), labeled P) fractions were analyzed by immunoblotting using anti-panGFAP (red channel) and anti-actin (blue channel) antibodies, which were used as a loading control. Quantification of GFAP mutants in the supernatant and pellet fractions was compared to WT controls (**H**). Data are mean ± SD, with each white dot representing a biological replicate (*n* = 3). Statistical significance was analyzed by two-tail *t*-test. * *p* < 0.05, ** *p* < 0.01.

**Figure 8 ijms-26-01913-f008:**
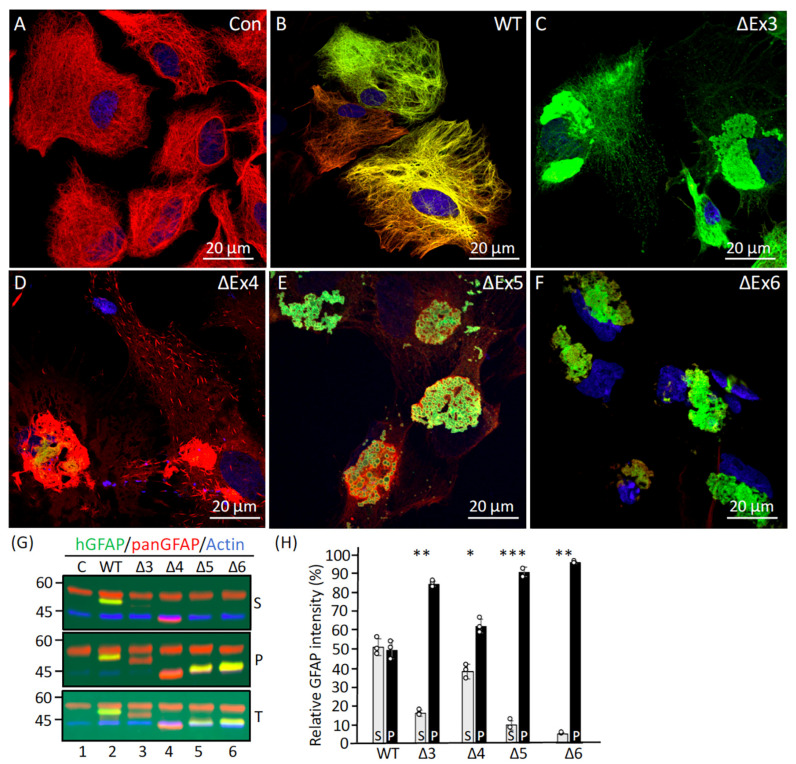
Filament organization and solubility properties of the internal truncated GFAP mutants in primary astrocytes. Primary astrocytes derived from WT rats (**A**) transduced with indicated GFAP expression constructs (**B**–**F**) were processed for immunofluorescence microscopy using anti-human GFAP (green channel) that preferentially recognized transduced human GFAP, and anti-panGFAP (red channel) antibodies that recognized both human GFAP and the endogenous rat GFAP. Nuclei were visualized by staining with DAPI (blue channel). Merged images were shown. Scale bar represents 20 μm (**A**–**F**). Note that astrocytes transduced with ΔEx3 (**C**) and ΔEx4 (**D**) GFAP produced diminished staining or no staining by anti-hGFAP antibody, as the epitope recognized by this antibody was missing because of the deletions. Thus, cells transduced with ΔEx3 mutant were stained with the GA-5 anti-GFAP antibody ((**C**), green channel), which recognized the C-terminal tail of GFAP. To assess biochemically the solubility properties of GFAP, astrocytes transduced with indicated GFAP expression constructs were extracted and the resulting supernatant ((**G**), labeled S) and pellet ((**G**), labeled P) fractions were analyzed by immunoblotting using anti-hGFAP (green channel) and anti-panGFAP (red channel) antibodies. Total cell lysates ((**G**), labeled T) were also analyzed by immunoblotting to assess GFAP expression levels in transduced cells. Immunoblots probed with anti-actin ((**G**), blue channel) were used as a loading control. Quantification of GFAP mutants in the supernatant and pellet fractions was compared to WT controls (**H**). Data were mean ± SD, with each white dot representing a biological replicate (*n* = 3). Statistical significance was analyzed by two-tail *t*-test. * *p* < 0.05, ** *p* < 0.01, *** *p* < 0.001.

**Figure 9 ijms-26-01913-f009:**
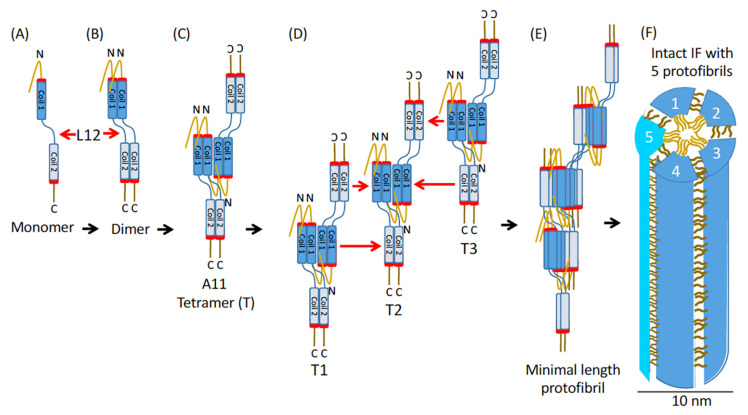
Schematic representation of the current model of IF assembly. This model, based on cryo-electron tomography and cryo-electron microscopy data [12], provides an important advance in our understanding of the architecture and assembly of vimentin filaments, with potential implications for GFAP as well. The accompanying diagram offers valuable insights into the progression of GFAP assembly, illustrating its transition from monomers (**A**) to dimers (**B**), tetramers, protofibrils (**C**–**E**), and ultimately mature filaments (**F**). Intact vimentin IFs consist of five protofibrils, each formed through successive interactions of three tetramers ((**D**), T1–T3) via three primary dimeric interaction configurations: A11, A22, and A12. The A11 tetramer is formed through dimeric interactions between the coil 1 region, while the A22 tetramer arises from interactions between the coil 2 region. The A12 tetramer is formed by interactions between the coil 1 and coil 2 regions. The T2 tetramer, aligned in the A11 configuration, acts as the core building block for unit length protofibril assembly. To complete the protofibril, the coil 2 dimers of the T1 and T3 tetramers align with the coil 1 tetramer of T2 (**D**). Additionally, the three tetramers are arranged so that their head domains ((**E**), depicted as dark yellow lines) form a fiber within the central lumen of the vimentin filament (**F**). The tail domains (depicted as dark blue lines) connect the protofibrils laterally through individual contact sites (**F**), ensuring a consistent filament diameter and radial compaction. In the intact 10-nm filament (**F**), the molecular interlock region ((**A**–**E**), highlighted by red bars) provides stability through conserved residue interactions between the N-terminus of coil 1A and C-terminus of coil 2B, which maintains the correct octameric structure at the central cross-section of the filaments. This assembly process is further facilitated by the flexibility of the L12 linker domain ((**A**,**B**), arrows), which enables coil 2 to align with coil 1 between incoming tetramers (**D**).

**Table 1 ijms-26-01913-t001:** Summary of the assembly efficiency of GFAP deletion mutants.

Deletion Variants	^1^ Assembly Efficiency	Significance (*p*-Value)
WT	++++	^2^ n.a.
NΔ42	+	very high (*p* < 0.01)
NΔ74	+	very high (*p* < 0.01)
NΔ118	+	very high (*p* < 0.01)
NΔ178	+	very high (*p* < 0.01)
NΔ225	+	very high (*p* < 0.01)
CΔ92	++++	^3^ n.d.
CΔ121	+++++	^3^ n.d.
CΔ207	+++++	^3^ n.d.
∆Ex4	++++	not significant
∆Ex5	+++++	high (*p* < 0.05)
∆Ex5 (50%)	+++++	high (*p* < 0.05)
∆Ex5 (25%)	++++	not significant

^1^ Assembly efficiency was measured after in vitro assembly by high-speed sedimentation assay as described in the Section 4. Quantification (+ to +++++) denotes the amount of GFAP detected in the pellet fraction. +++++: >80% GFAP in the pellet; ++++: 60–80% GFAP in the pellet; +: <20% GFAP in the pellet. Statistical significance was evaluated using a two-tailed *t*-test. *p* values are indicated. Quantification results were obtained from three independent experiments for each deletion mutant, except for ∆Ex4, which was based on two independent experiments. ^2^ n.a., not applicable. ^3^ n.d., not determined. Due to their high aggregation causing precipitation, these deletion mutants were analyzed by low-speed sedimentation to assess filament aggregation instead.

**Table 2 ijms-26-01913-t002:** Summary of deletion mutant phenotypes.

Deletion Variants	Deletion Location	Major Phenotype	^1^ Length (nm)	^2^ Width (nm)	GFAP in Primary Astrocyte	
					Phenotypes	Extractability
WT	no	10 nm filaments	>2000	10.2 ± 1.1	Filamentous networks	^3^ High
NΔ42	head domain	abnormal filaments	516.72 ± 193	17.33 ± 5	n.d.	n.d.
NΔ74	head+1/4 coil 1A	short fibrous structures	235.89 ± 81	16.75 ± 5	n.d.	n.d.
NΔ118	head+coil 1A	unassembled materials	108.61 ± 53	17.08 ± 2.5	n.d.	n.d.
NΔ178	head+coil 1A+2/3 coil 1B	unassembled materials	109.95 ± 62	25.87 ± 9	n.d.	n.d.
NΔ225	head+coil 1A+coil 1B	unassembled materials	108.33 ± 54	50 ± 25	n.d.	n.d.
CΔ92	tail+1/3 coil 2B	abnormal filaments	736.91 ± 353	26.72 ± 9	cytoplasmic aggregates	n.d.
CΔ121	tail+2/3 coil 2B	paracrystal aggregates	n.a	n.a	cytoplasmic aggregates	n.d.
CΔ207	tail+coil 2A + coil 2B	amorphous aggregates	n.a	n.a	cytoplasmic aggregates	n.d.
∆Ex3	2nd half of coil 1B	roundish aggregates	219.37 ± 153	80.63 ± 25	cytoplasmic aggregates	^5^ Low
∆Ex4	coil 1B + coil 2A	abnormal filaments	571.46 ± 237	23.82 ± 7	needle-like structures	^4^ Medium
∆Ex5 *	1st half of coil 2B	filamentous aggregates	295.63 ± 143	16.46 ± 5	Ring-like deposits	^5^ Low
∆Ex6	2nd half of coil 2B	rod-like aggregates	145.65 ± 73	35.22 ± 10	cytoplasmic aggregates	^5^ Low

^1^ Filament lengths were estimated from representative micrographs and expressed as mean ± SD. Where the average filament length was especially long, it was then expressed as greater than the length that could be accurately measured. ^2^ Single filamentous structures were used for width measurements, obvious bundled or unraveled filaments were not included; ^3^ High: >75% GFAP in the pellet; ^4^ Medium: 25–75% GFAP in the supernatant; ^5^ Low: <25% GFAP in the supernatant. n.d., not determined; n.a., not applicable. * Filaments were measured at the periphery of large aggregates of GFAP.

**Table 3 ijms-26-01913-t003:** Primary antibodies used for immunostaining and immunoblotting.

Antibody (Clone No.)	RRID	Host	Assay Dilution	Supplier
GFAP (SMI21)	AB_509978	Mouse	IB: 1:5000 IF: 1:500	BioLegend (San Diego, CA, USA)
GFAP-α	AB_10672298	Mouse	IB: 1:5000 IF: 1:500	NeuroMab (Davis, CA, USA)
GFAP	AB_2631098	Rabbit	IB: 1:5000 IF: 1:500	Cell signaling Technology (Danvers, MA, USA)
GFAP	AB_10013482	Rabbit	IB: 1:10,000 IF: 1:1000	Dako (Santa Clara, CA, USA)
GFAP (GA5)	AB_721051	Mouse	IB: 1:5000 IF: 1:500	Sigma-Aldrich
Vimentin (V9)	AB_609914	Mouse	IB: 1:5000 IF: 1:500	Sigma-Aldrich
Vimentin	AB_10695459	Rabbit	IB: 1:5000 IF: 1:500	Cell Signaling Technology
Actin	AB_787885	Mouse	IB: 1:5000	Novus (Centennial, CO, USA)
Actin (Rhodamine-conjugated)	AB_2861334	Mouse	IB: 1:10,000	Bio-Rad
Aquaporin 4	AB_91530	Rabbit	IF: 1:200	Sigma-Aldrich

IB: immunoblotting; IF: immunofluorescence.

## Data Availability

The data that support the findings of this study are available from the corresponding author upon reasonable request.

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
