# Peer review of "Deletions in Glial Fibrillary Acidic Protein Leading to Alterations in Intermediate Filament Assembly and Network Formation"

_ijms, 2025, doi:10.3390/ijms26051913_

Round 1
Reviewer 1 Report
Comments and Suggestions for Authors
The manuscript titled "Deletions in Glial Fibrillary Acidic Protein Leading to Alterations in Intermediate Filament Assembly and Network Formation" provides a comprehensive investigation into the molecular effects of GFAP deletions on filament assembly and organization, advancing our understanding of intermediate filaments in the context of Alexander disease. The authors have designed well-structured experiments, presented robust data, and identified key regions of GFAP critical for filament formation and network stability. However, the manuscript could benefit from further clarification in certain areas, such as a quantitative comparison of assembly efficiency and a more detailed summary of the mutant phenotypes. Additionally, a broader discussion on the therapeutic implications of these findings would enhance the manuscript’s relevance. Specific suggestions and questions for further refinement are provided in the attached file.

Reviewer 2 Report
Comments and Suggestions for Authors
This is a very thorough and clear study on delineating the role of the sequences within GFAB protein in filament assembly and is important in understanding diseases associated with mutations in GFAB and Desmin and other type IiI filament proteins. The work is very well done. I have only minor corrections for some of the english and a few minor issues with figures, detailed below:
1. Line 40, delete comma after domain and add “is” after which, making it …which is flanked by …… add “a” after by
2. Line 57-58: change “in the presence of physiological amounts of salt and pH” in solutions mimicking physiological pH and salt concentrations”
3. Line 61 change UFL to ULF
4. Line 66: change study to studies or add an “a” after upon in line 65
5. On line 77 add an “s” at the end of extension
6. On line 99 change “have facilitated” to “provide”
7. In Figure 2 please make the Y-axis labels in D and larger font and provide a Y-axis label for figure 2F
8. Line 168 add an “s” at the end of “leave”
9. Line 198 change “previously” to “previous”
10. Figure 6 the label for the Figure C appears to be I, please correct or explain.
11. On line 477 it says “PR box” then on line 478 it says RP-box please fix this discrepancy.
12. Fix spacing for lines 613-615
Reviewer 3 Report
Comments and Suggestions for Authors
Review on manuscript: ijms-3387539
Deletions in glial fibrillary acidic protein leading to alterations in intermediate filament assembly and network formation
by Ni-Hsuan Lin, Wan-Syuan Jian, Ming-Der Perng*
submitted to IJMS
Research paper
This manuscript investigated that how deletions in glial fibrillary acidic protein lead to the alterations in intermediate filament assembly and network formation. Overall, it is interesting and written very well. Also, the authors provided sufficient research data to support their findings. Hence, I just provide some suggestions and comments about the structure and writing of this manuscript to further improve its whole quality, which have been shown as follows.
Detailed recommandations:
-Materials and Methods: strongly suggest to change the first paragraph to “4.1. Materials”, which is used to list the reagents, consumable items, and special materials utilized in this study. In addition, their sources, suppliers, and/or purities should also be presented.
-Table 1: it should be shown in a normal three-line form.
-Statistical analysis: How many biological replicates did the authors perform for each experiment? The information should be presented herein or in the stated when describe each method.
-Lines 164-165: please check the sentence.
-References: some cited publications were too old, which can be changed to mostly recent publications.
-Figgure 3A-F: what's the length for each scale bar representing? This information is missing.
-Figure 5A-F: Idem.
-Figure 7A-F: Idem.
-Figure 8A-F: Idem.
Reviewer 4 Report
Comments and Suggestions for Authors
This MS is about construction of GFAP deletion mutants and testing for their effects on filament formation. The cryoEM (+ ET + FIB) structure of a closely related protein vimentin has been published, and so have the crystal structures of fragment 1B of GFAP and other fragments of vimentin. The known structures should provide direct explanation for the outcome of the deletion mutants. When briefly reviewing the structures, the authors can also include a schematic diagram showing how the type III IF proteins assemble into tetramers and pentamers in the intermediate filaments.
There are isoforms of GFAP due to alternative splicing. The authors did not indicate which isoform they used in this study.
In the schematic diagram of Figure 1 the authors should mark the locations of the nine exons in GFAP. The latter four mutants should be renamed as ΔEx3, 4, 5 and 6 for consistency with the description in Section 2.4 and elsewhere. When discussing the mutant results in the coiled coil regions, the heptad repeat and the length of deletion should be taken into consideration.
In Figure 6A, sequence alignment of the entire region of exons 3, 4, 5 and 6 should be presented along with the location of each exon as well as the helical structure from the cryoEM and Xray studies.
The first two paragraphs of Section 3 need a heading. Otherwise it is hard to grasp what the authors would like to address. Why is DNA (or RNA) involved here?
Section 5 can be merged with Section 3 as the final subsection of discussion (under a heading like “concluding remarks”).
Page 16, line 477 and elsewhere: Is it PR box or RP box?
Reviewer 5 Report
Comments and Suggestions for Authors
In this manuscript, the authors have investigated the effects of the sequences of GFAP on its intermediate filament assembly by applying deletion mutants. They have revealed that deletions at the C-terminal end result in abnormalities in either filament diameter calibration or lateral association, whereas deletions at the N-terminus significantly disrupt the filament assembly process. In addition, the filament-forming sequences within the rod domain varied in their contributions to filament assembly and network formation.
Below, my main specific comments are provided:
1. In this manuscript, deletions have been applied to assess the effects of sequence change on the protein assembly (filament/aggregate formation). However, the backbones are vitally important to the protein structure and filament/aggregate assembly. Deletion of a sequence especially within a domain region or linker may inevitably disrupt the structure and assembly properties. So, multiple-point mutations in some key residues should be employed in this study.
2. I understand that the authors have used both high-speed and low-speed sedimentation to fractionalize filaments (Fig. 3G,F; Fig. 5H) and aggregates (Fig. 4D, Fig. 5G), but I wonder whether the filament/aggregate conversions occurred during the high/low-speed sedimentation, and which sedimentation speed should be selected for an assembling stage (amplitude).
3. The authors have mixed the deletion mutant with WT GFAP with indicated ratios but with different concentrations. As known, protein aggregation is dependent on its concentration, different concentrations may lead to different aggregation propensities. It is suggestive to keep a same concentration of the deletion mutant and the increasing concentration of the WT to get variable ratios.
4. It seems that the deletion mutants are prone to aggregation and sequestering the WT GFAP into aggregates. This dominant-negative effect may often occur when a mutant is over-expressed in cells and sequesters the WT protein into aggregates. This phenomenon should be described clearly and discussed in detail.
Round 2
Reviewer 5 Report
Comments and Suggestions for Authors
This revision has addressed my concerns partly, and it has described and discussed clearly the dominant-negative effect related to the mutants (#4). I understand that the authors are not possible to complete the experiments on point mutations (#1) and WT/Mutant ratios in a constant concentration of the Mutant (#3) within a short revising period.
I would like to recommend publishing the manuscript in the present version.